# How Immunonutritional Markers Are Associated with Age, Sex, Body Mass Index and the Most Common Chronic Diseases in the Hospitalized Geriatric Population—A Cross Sectional Study

**DOI:** 10.3390/nu16152464

**Published:** 2024-07-29

**Authors:** Serena S. Stephenson, Ganna Kravchenko, Renata Korycka-Błoch, Tomasz Kostka, Bartłomiej K. Sołtysik

**Affiliations:** Department of Geriatrics, Healthy Ageing Research Centre (HARC), Medical University of Lodz, Haller Sqr. No. 1, 90-647 Lodz, Poland; serena.stephenson@umed.lodz.pl (S.S.S.); ganna.kravchenko@umed.lodz.pl (G.K.);

**Keywords:** immunonutritional markers, inflammaging, immunosenescence, older, concomitant diseases

## Abstract

The aim of this study was to assess the relationship of different chronic diseases with immunonutritional markers in the senior population. Methods: this study included 1190 hospitalized geriatric patients. The criteria to participate were ability to communicate, given consent and C-reactive protein (CRP) lower than 6 mg/dL. Results: the mean age of the study population was 81.7 ± 7.6 years. NLR (neutrophil-to-lymphocyte ratio), LMR (lymphocyte-to-monocyte ratio), MWR (monocyte-to-white blood cell ratio), SII (systemic immune–inflammation index), PNI (prognostic nutritional index) and CAR (C-reactive protein-to-albumin ratio) were related to age. NLR and MWR were higher, while LMR, PLR (platelet-to-lymphocyte ratio and SII were lower in men. All markers were related to BMI. NLR, LMR, LCR (lymphocyte-to-CRP ratio), MWR, PNI and CAR were related to several concomitant chronic diseases. In multivariate analyses, age and BMI were selected as independent predictors of all studied immunonutritional markers. Atrial fibrillation, diabetes mellitus and dementia appear most often in the models. PNI presented the most consistent statistical association with age, BMI and concomitant chronic diseases. Conclusions: this study reveals the pivotal role of aging and BMI in inflammatory marker levels and the association of immunonutritional markers with different chronic diseases. Atrial fibrillation seems to have the most dominant connection to the immunonutritional markers.

## 1. Introduction

The ageing process is likely to bring about substantial remodeling of the immune system, referred to as immunosenescence, and heightened systemic inflammation, known as inflammaging. Both factors contribute to an elevated risk of developing and intensifying chronic diseases in old age [1]. Human ageing is additionally linked to a gradual decline in immune functions and an increased susceptibility to infections. Nevertheless, establishing goals or priorities for older patients dealing with chronic health conditions or multimorbidity is a complex task. The conventional disease-specific guidelines often do not readily apply to senior individuals managing multiple, overlapping and age-related health conditions [2].

Immunonutrition is an emergent and interdisciplinary subject, since it comprises various aspects related to nutrition, immunity, infections, inflammation, injury or tissue damage [3]. Several studies have been performed regarding the development of the best biomarkers, useful for identifying patients who could benefit from the prediction of treatment by chemotherapy, chemoradiotherapy and preoperative surgeries [4,5,6]. Several inflammatory markers including levels of neutrophils, albumin, C-reactive protein (CRP), lymphocytes, platelets, and the combination of certain ratios have been studied for prediction of diseases such as coronary heart disease [7] and neoplasms [8]. Using these easily available morphological and biochemical parameters, various inflammation-derived markers can be calculated. The laboratory tests mentioned supply insights into nutritional status (lymphocytes, albumin), immunological status (white blood cells, platelets), and inflammation (CRP, leukocytes). The integration of these variables enables a more comprehensive assessment of immunonutritional decline [3,5].

Neutrophil-to-lymphocyte ratio (NLR) and lymphocyte-to-monocyte ratio (LMR) are hailed as novel markers for inflammation. Their levels are linked to overall mortality and severity of diseases, and may show potential adverse effects on certain patient conditions [9]. An elevated level of the recently studied inflammation marker, NLR, has been found to be correlated with the onset of frailty [7] and neoplastic diseases [10]. Similarly, LMR has been implicated in pulmonary embolism [11] and neoplastic conditions [12]. Importantly, the combination of certain inflammatory markers such as the platelet-to-lymphocyte ratio (PLR) is acknowledged as a valuable biomarker predicting mortality within one month after sepsis [13]. This marker proves its utility in neoplasms [14], heart failure [15] or rheumatic diseases [16] and is considered to be a general marker of overall health [17]. Lymphocyte-to-C-reactive protein ratio (LCR) has been utilized for the prognosis of certain cancer diseases [12,18], and it is also referenced in the context of COVID-19 infection [19]. This marker combines the assessment of the balance between inflammation and nutrition [20]. Another biomarker—monocyte-to-white blood cell ratio (MWR)—is associated with certain health conditions, particularly those involving inflammation or immune system dysregulation. Among notable disorders are inflammatory deficiencies, infections and cardiovascular diseases, as monocyte activation may be linked with some cardiovascular diseases or atherosclerosis [21]. The systemic immune-inflammation index (SII) has been associated with the prognosis of many cancer types. SII served as a strong indicator of tumor differentiation [22] as well as to predict survival outcome in patients with gastric cancer [23]. Moreover, SII collectively was used to predict hospital mortality in COVID-19 patients and may ease early risk evaluation [24]. The Prognostic Nutritional Index (PNI) acts as a nutritional indicator and predictor of various diseases. PNI has been remarkably associated with a higher incidence of ischemic stroke in patients with non-cardiac surgery [25]. In older subjects with the risk of chronic obstructive pulmonary disease exacerbation and chronic kidney disease, PNI is considered to be useful in monitoring nutritional status in this population [26,27]. C-reactive protein-to-albumin ratio (CAR), has been suggested as an effectively utilized inflammatory marker [28,29]. CAR is recognized to be a reliable and applicable tool for detecting and finding the clinical severity of acute severe ulcerative colitis [30].

To the best of our knowledge, there are no studies exploring comprehensively the spectrum of immunonutritional markers not in the context of cancer or inflammatory diseases but within various chronic conditions prevalent in the older multimorbid population. The findings from such research would provide insights into immunosenescence processes stemming from the most prevalent health issues in this age-group. Therefore, the aim of the current study was to explore the association of inflammatory markers (NLR, LMR, PLR, LCR, MWR, SII, PNI and CAR) with age, sex, BMI and concomitant diseases and to identify the factors potentially influencing nutritional and immunological status in older adults’ hospitalized population.

## 2. Materials and Methods

### 2.1. Study Design

We evaluated patients admitted to the Geriatric Department, Central Veterans Hospital located in Lodz, Poland. In this cross-sectional study, the population were older adults, aged 60 years old and above. Patients were recruited from January 2012 to September 2023. From 2020–2022, the Department of Geriatrics served partially as a COVID-19 ward. The initial total number of individuals selected was 4600 patients. As the study’s assumption was to examine individuals without a current heightened inflammatory state, 2400 patients were excluded due to CRP higher than 6 mg/dL. Furthermore, 1010 patients were eliminated due to lack of complete laboratory or anamnesis data. Patients were included in this study with the following inclusion criteria: admission to the department, aged 60 years and above, complete data, and giving informed consent. Consecutive admissions of the same individuals from over the years were not included in this analysis. After screening, 1190 patients (883 women and 307 men) who met the criteria were enrolled into the analysis (Figure 1). All the participants were subjects with non-infectious diseases and usually several concomitant diagnoses (e.g., heart failure, coronary heart disease, dementia, depression, anemia, osteoporosis, chronic kidney disease, pulmonary disorders, gastrointestinal disorders, falls, fractures and diabetes mellitus).

### 2.2. Data Collection and Laboratory Tests

Data collection for the study involved obtaining information on age and sex. Measurements of body mass and height were taken with individuals barefooted, enabling the calculation of Body Mass Index (BMI). Laboratory parameters, including a full blood count (white blood cell count, neutrophils, monocytes, lymphocytes, and platelets), were analyzed using the Sysmex XN 2000 analyzer (Kobe, Japan). Additionally, CRP levels and albumin concentration were determined using the Beckman Coulter Dx700 AU analyzer (Brea, CA, USA). All laboratory parameters were obtained upon patient admission.

Immunonutritional biomarkers were evaluated as follows: NLR is determined by dividing the absolute neutrophil count by the absolute lymphocyte count in a blood sample [17]. LMR is assessed by dividing the absolute lymphocyte count by the absolute monocyte count in a blood sample [11]. PLR is calculated by dividing the absolute platelet count by the absolute lymphocyte count in a blood sample [15]. LCR is computed by dividing the absolute lymphocyte count by the CRP [mg/L] in a blood sample [18]. MWR is calculated by dividing the absolute monocyte count by the total white blood cell count in a blood sample [31].The Systemic Immune–Inflammatory Index is represented by a numeric value derived from peripheral blood counts, specifically the platelet count, neutrophil count, and lymphocyte count. It is calculated using the formula (Platelet count × Neutrophil count)/Lymphocyte count [17]. The prognostic nutritional index (PNI) is calculated with the equation of PNI = 10×serum albumin (g/dL) + 0.005 × total lymphocyte count (per mm^3)^ [25]. The CRP-to-albumin ratio (CAR) is determined by dividing CRP [mg/L] by albumin [g/dL] [29] (Figure 2).

### 2.3. Concomitant Diseases

Diseases such as arterial hypertension, diabetes, lipid disorders, stroke, coronary artery disease, myocardial infarction, arterial fibrillation, heart failure, chronic kidney disease (CKD), obstructive lung disease (COPD and asthma), osteoarthritis, osteoporosis, fractures, gastrointestinal diseases (chronic gastritis, gastrointestinal ulcer), neoplastic diseases, depression, dementia, Parkinson’s disease, pressure ulcer and urine incontinence were analyzed. CKD was defined as a glomerular filtration rate (GFR) lower than 60 mL/min/1.73 m^2^ according to the BIS1 formula [32]. Depression was assessed as previously diagnosed and/or by using the Geriatric Depression Scale [33], where a score of five or higher indicates the presence of depression. Dementia was diagnosed based on the results of the Mini-Mental State examination, where a score of less than 24 points indicated the presence of dementia [34].

### 2.4. Statistical Analysis

The normal distribution was verified with the Shapiro–Wilk test. As several variables were not normally distributed, data were expressed both as the mean ± standard deviation (SD) and median (quartiles). Quantitative variables were compared using the Mann–Whitney U-test due to the non-normal distribution and lack of homogeneity of variance. Qualitative variables were assessed using a Chi-square test. Spearman correlation coefficients were used to calculate the relationship between the quantitative variables. To identify the immune marker responsible for the increased risk of chronic diseases in our study, we follow a 2-step approach. First, comparisons between qualitative variable calculations were performed using the Mann–Whitney test. Second, general linear models were used for statistically significant data in bivariate models. Since the values of immunonutritional markers were not normally distributed, they were logarithmically transformed, and obtained values were reused in general linear models. Statistical significance was set at *p* ≤ 0.05. Statistical analysis was performed using Statistica 13.1.

### 2.5. Ethical Certification

This study was conducted according to the guidelines of the Declaration of Helsinki and approved by the Ethics Committee of the Medical University of Lodz with the approval number (RNN/67/23/KE). Patients signed informed consent for all the diagnostic and therapeutic procedures during hospitalization. All the data gathered were confidential.

## 3. Results

The mean age of the study population was 81.65 ± 7.57 years. Table 1 shows the characteristics of 1190 patients according to sex. In terms of immunonutritional markers, women had significantly higher LMR, PLR, SII and lower MWR. In the tested population, women were significantly older than men. Furthermore, women presented with significant frequency arterial hypertension, lipid disorder, heart failure, osteoarthritis, osteoporosis, incidence of fractures, depression, dementia and urine incontinence. In contrast, the prevalence of myocardial infarction, gastrointestinal diseases, neoplasms and Parkinson’s disease was higher the in group of men.

In Table 2, we present the correlations (rho and *p* values) for the most important quantitative variables, such as age and BMI. Calculations were performed for the whole study group and by division of sex.

Considering that most of the studied biomarkers exhibited correlations with age, Figure 3 includes matrices depicting them. Specifically, that LMR and PNI exhibited a negative correlation, while NLR, LMR, MWR, SII and CAR demonstrated a significant increase with age. PLR and LCR were not correlated with age.

In Table 3, we present the results of the bivariate Mann-Whitney U-tests for immunonutritional markers with regard to the tested diseases. The table comprises only those diseases, which were associated with significant distinct levels of immunonutritional markers. The three conditions expressed only outlying associations with, and have not been placed in Table 3. PNI for subjects with coronary artery disease for the whole group was significantly lower z = −2.08 *p* = 0.03 in comparison with those without this disease. Men with a history of neoplasms displayed significantly lower PNI z = −2.13 *p* = 0.03. The ratio was significantly lower in past-fracture groups z = −3.62, *p* < 0.001. In division by sex, the association was present in the group of women z = −3.28, *p* = 0.001. The diseases such as arterial hypertension, myocardial infarction and depression expressed no significance in bivariate models.

Following the bivariate analysis, we utilized general linear models to investigate the factors affecting the studied immunonutritional markers. The model-finding procedure followed a stepwise backward approach, where initially all variables showing significance in the bivariate analysis were included, and then insignificant independent variables were systematically eliminated. The final models were formed with only those variables that had statistical significance.

For NLR in the final model, only age was significantly associated. In the multivariable test, LMR was influenced by age, heart failure and atrial fibrillation. PLR was negatively related to BMI, presence of atrial fibrillation, and male sex. LCR and SII were linked to BMI. MWR presented an association with age, male sex, diabetes mellitus, atrial fibrillation and dementia. PNI was negatively influenced by age, previous stroke, atrial fibrillation, and dementia, and positively by lipid disorders and osteoarthritis. In the case of CAR, a positive impact was revealed for age, BMI, diabetes mellitus, and atrial fibrillation, and a negative one for osteoporosis. The equations of the models can be found in Table 4. As the values of immunonutritional markers did not meet the criterion of normal distribution, the data were logarithmized and used to estimate generalized linear models. However, the components of the models did not differ from those estimated for raw data.

## 4. Discussion

To the best of our knowledge, the current study provides the first examination of the relationship of different chronic conditions with immunonutritional markers within a geriatric population. Our findings suggest that diverse diseases can potentially alter the examined markers, yet the data presented suggest a relatively consistent pattern in terms of a predominant relationship with BMI and age.

As a marker combining both inflammatory response and indirect nutritional status, NLR is acknowledged as a good predictor of the outcome in multiple disorders, such as intracerebral hemorrhage [35], neoplasms [36], heart failure [37] or schizophrenia [38]. In the presented study, NLR was positively associated with age, diabetes mellitus, chronic heart failure and chronic kidney disease, but negatively with BMI, lipid disorders and gastrointestinal diseases. The multivariable test, however, indicates that NLR may be predominantly connected with age. The difference may result from the uniqueness of the tested group. For the nongeriatric population, the mean of NLR ranges around 1.5–2 [39,40], and the mean in our study is 3.18. The positive correlation between age and NLR is observed as well in the young healthy population [40]. The detected discrepancy may indicate the process of inflammaging and a progressive imbalance between neutrophils and lymphocytes. The lack of disorders in the multivariable model may be connected with the fact that NLR has been negatively correlated with BMI. There are papers presenting the association of NLR with malnutrition, but these studies refer to oncologic disorders [41] or obstructive lung diseases [42]. Malnutrition, and, predominantly, ageing seem to play a crucial role in changing the level of NLR in the examined population.

The predictive value of LMR is intrinsically connected with neoplasms [43,44], but the marker seems to be specific for other disorders such as COVID-19 infections [45], strokes [46,47] or kidney disease [48,49]. Our study indicates that lower LMR was associated with age, past strokes, atrial fibrillation, heart failure, chronic kidney disease and chronic lung disease. In contrast, a higher value of LMR was linked to higher BMI and presence of lipid disorders. The last interaction may be potentially explained by the fact that subjects with lipid disorders in our population were significantly younger. The multivariable model shows that LMR may be mainly negatively influenced by age, male sex, presence of heart failure, and atrial fibrillation. For heart failure, LMR can be considered as an independent biomarker of worse outcome [47]. The available literature mentions the association of low LMR with atrial fibrillation [50] and as a potential predictor of all-cause mortality [51]. A Chinese study indicates that the sex-dependent value of LMR, for both below- and above-65-year-old males had significantly lower LMR. In this paper, a consecutive decrease in the ratio with age is also observed [52]. It may be assumed that both heart failure and atrial fibrillation, regardless of age and sex, impact negatively on this immunonutritional marker among older people.

PLR, in our observation, was negatively associated with male sex, BMI, incidence of diabetes, and atrial fibrillation, and positively with the occurrence of osteoporosis. A multivariable calculation for PLR reveals a negative influence of male sex, BMI, and atrial fibrillation. There are some papers indicating no sex difference for PLR, but these studies were conducted among a significantly younger population [52,53]. Likewise, there was no difference in PLR between subjects with and without atrial fibrillation, but the study was conducted among the critical-care patients with a wide age range [54]. However, another study presents comparable results to ours, demonstrating lower PLR in atrial-fibrillation patients and concluding a possible deteriorating effect on atrial remodeling and thrombogenesis [55]. The interpretation of those data is not obvious. Some research reports a reverse correlation between BMI and PLR, but through the prism of neoplastic disease [56]. It seems that PLR, regardless of age, may reflect not only atherosclerosis, inflammation, and thrombocytes activation [57], but also worse nutritional status or, finally, increased all-cause mortality in the senior group [58].

LCR, known as a lymphocyte-to-CRP ratio, reflects the balance between nutritional and immunological status, and higher values reflect better prognosis. In the presented study, the marker was significantly higher in subjects with lipid disorders, osteoporosis and gastrointestinal diseases. Subjects diagnosed with these diseases were significantly younger, which may be partially explained by the less-intensive diagnosing in advanced-age subjects. A reverse worse ratio was observed when atrial fibrillation, heart failure and chronic kidney disease were present. Furthermore, LCR showed a negative correlation with BMI. Multivariate analysis has revealed the prevailing negative impact of BMI. Our research indicates that both lymphocytes and CRP correlate positively with BMI, but the increase in CRP is significantly higher than that of lymphocytes. This may be understood as the dominant proinflammatory effect of fatty tissue. In this regard, BMI seems to have a negative impact on LCR. In the available literature, LCR is acknowledged as an inflammation marker in patients with HIV infection [59] or neoplasms [18,20]. The potential of LCR may be considered, as well, for the disorders significant in bivariate calculations, such as atrial fibrillation, heart failure, and chronic kidney disease.

MWR (monocyte-to-white blood cell ratio) in bidirectional calculation was significantly lower in subjects with diabetes mellitus and dementia, but higher in atrial fibrillation, chronic heart failure, obstructive lung diseases, and gastrointestinal diseases. Men presented significantly higher MWR. Age was positively, and BMI negatively, corelated with MWR. Furthermore, in multivariable analysis, MWR was positively associated with age, male sex, and atrial fibrillation, and negatively with diabetes mellitus and dementia. In some papers, high MWR reveals the predictive value of prognosis of neoplasm disease [31,60]. No literature regarding the association between MWR and atrial fibrillation was discovered. However, there is evidence suggesting that a higher monocyte count is linked to a poorer prognosis in patients with atrial fibrillation, [61] and activation of these blood cells may contribute to a predisposition to arrythmias [62]. Additionally, an elevated white blood cell count is also correlated with an increased risk of atrial fibrillation [63,64]. The existing literature does not indicate a negative association between MWR and type 2 diabetes or dementia. Notably, type 2 diabetes is significantly linked to inflammation and elevated BMI, leading to an increased count of white blood cells, relative to monocytes. Dementia tends to occur more frequently with advancing age. While these considerations may elucidate, to some extent, our findings, a thorough and comprehensive analysis is necessary in future research to fully understand the relationship between age, diseases and MWR.

The systemic immune–inflammation index, which combines the count of platelets, neutrophils and lymphocytes in the presented study, was significantly lower in the atrial fibrillation group. Furthermore, the index has a negative correlation with BMI, and a positive one with age. In a multivariable test, SII was influenced only by BMI. The research holding over forty thousand subjects acclaimed SII as a marker which is strongly associated with cardiovascular death and concomitantly indicates systemic inflammation [65]. The evidence from the NHANES study provides data showing that higher SII is associated with diabetic kidney disease [66], osteoporosis [67] or hyperlipidemia [68]. SII may be also acknowledged as a useful marker in oncology [22,69]. Our data show that SII may be considered as the marker of immunosenescence. Negative correlation with BMI suggests that SII is combined with nutritional status. The association with concomitant diseases was rather weak. Surprisingly, our participants with atrial fibrillation exhibited lower SII. Contrary to expectations, considering that atrial fibrillation is typically associated with poorer immunonutritional outcomes, this result can be attributed to neither the potential anti-inflammatory effects of anticoagulant therapy nor the younger age of individuals with atrial fibrillation—here, the correlation is the opposite. This aspect requires more research for a thorough understanding.

The prognostic nutritional index is a marker strongly associated with malnutrition, both in neoplasms [70,71] and in chronic diseases such as heart failure [72]. On the other hand, according to some authors, PNI is more associated with inflammation than nutrition [73]. In our material, PNI was significantly lower in patients with history of stroke, atrial fibrillation, coronary artery disease, chronic heart failure, and chronic kidney disease, and a history of fractures, neoplasms, dementia, and was and higher in subjects with lipid disorders and osteoarthritis. PNI also expressed a positive correlation with BMI and a negative one with age. A multivariable model has shown that PNI was influenced negatively by age, presence of stroke, atrial fibrillation and dementia, and positively by lipid disorders and osteoarthritis. In the available literature, there are papers indicating an association of worse PNI in diabetic subjects [74,75] or in kidney failure [75,76]. In terms of stroke, there is some research indicating on prognostic value of PNI in acute phase of ischemic stroke [77] or in perioperative ischemic stroke-risk prediction [25]. Our work in this field indicates the association between worse PNI and incidence of a previous cerebrovascular event. In terms of atrial fibrillation, a low value of PNI is associated with a significantly increased risk of death [78]. Apparently, there are no papers indicating the association between dementia, osteoarthritis, lipid disorders, and PNI. However, those chronic disorders are strongly connected with nutritional status, and people with dementia tend to develop malnutrition, as opposed to lipid disorders and osteoarthritis, which usually apply to younger well-nourished people with a higher BMI. In this context, PNI appears to be, among others, the most discriminative indicator with effectiveness in assessing various factors such as prevalent diseases, age, and nutritional status.

Elevated CRP levels have been associated with major cardiovascular events in individuals with coronary artery disease and atherosclerosis burden [28]. Additionally, the serum CRP/albumin ratio has been proven to be a more precise marker for predicting the prognosis of critical diseases, compared to individual albumin and CRP levels [29]. In our study, bivariate analysis revealed a strong positive association between CAR and diabetes mellitus, atrial fibrillation, heart failure, and chronic kidney disease, but a negative one for osteoporosis and gastrointestinal diseases. CAR was positively correlated with age and BMI. Multivariable analysis showed that CAR is positively influenced by age, BMI, and the presence of diabetes mellitus and atrial fibrillation, but negatively by osteoporosis. Numerous studies highlight the broad usefulness of CAR in acute stages [79,80] or neoplasms [81,82]. However, there is a current lack of studies indicating a potential positive association between CAR and BMI. It is well known that fatty tissue triggers inflammation, as already proven for polycystic ovary syndrome [83], but for the older population, the correlation still requires confirmation. Another noteworthy relationship is the simultaneous increase in age and CAR, which may signify immunosenescence, regardless of other covariates. Existing research has already identified CAR as a predictor of postoperative atrial fibrillation after coronary artery procedures [84], or during COVID-19 infection [85]. Our data suggest a robust independent association between elevated CAR and the presence of atrial fibrillation. The impact of this association may be bidirectional, as heightened CAR could contribute to the initiation of atrial fibrillation. Conversely, atrial fibrillation, through impaired circulatory function, and, indirectly, malnutrition and inflammation, may elevate CAR. The level of CAR is proven to be significantly higher among subjects with complications during the course of diabetes mellitus [86]. Nevertheless, our data directly indicate that diabetes mellitus itself, likely due to hyperglycemia and hyperinsulinemia, may raise CAR.

Several general observations made in the present study should be highlighted. All the assessed markers were generally age- and BMI-dependent. The majority were related to age and BMI in an opposite way, indicating a deleterious impact of age and a protective role for BMI. LCR and CAR behaved differently: the relationship with age and BMI went in the same direction. This probably reflects the combined influence of nutritional and inflammatory components, with CRP outweighing the effects of nutritional factors. While BMI seems to have a protectional impact on nutritional components of markers, it is directly related to CRP in a generally linear manner. Therefore, this “interplay” of nutritional and inflammatory components results in a unidirectional association of age and BMI with LCR and CAR. Finally, the power of different markers to relate to clinical conditions seems to vary significantly, with PNI being the most sensitive one.

One of the major strengths of this research lies in the deliberate removal of subjects with elevated CRP levels, resulting in a more homogenous study population in terms of inflammation. This strategic approach offers a unique opportunity to gain new insights into the immunonutritional balance among the older people. Additionally, a noteworthy strength is the inclusion of a significantly large study population. However, it is essential to acknowledge several limitations inherent in this study. Firstly, the focus was on older inpatients in central Poland, and while these patients presented with multiple medical problems, patients in other regions and cultures may have different comorbidity profiles. Furthermore, the study is of an observational nature, which emphasizes the need for caution in drawing causal relationships. To overcome these limitations and enhance the generalizability of findings, future research endeavors could benefit from larger multicenter studies conducted in diverse populations, easing more comprehensive and nuanced conclusions.

## 5. Conclusions

Our study sheds light on the significance of a specific set of biomarkers within the framework of ageing and the presence of concurrent chronic diseases. Age and BMI stand out as the fundamental and crucial variables influencing the levels of immunonutritional markers. Sex is also noteworthy, as men exhibit considerably lower LMR, PLR, and MWR. The presence of concurrent diseases is significant, particularly age-related conditions like atrial fibrillation, heart failure, previous stroke and chronic kidney disease, related to the concentrations of the mentioned biomarkers. Conditions occurring in younger individuals, such as lipid disorders or osteoarthritis, relate generally in an opposite direction, as compared to the typical diseases of advanced age. PNI appears to have the best discrimination capacity for prevalent diseases, age, and nutritional status. Our study underscores the practicality of easily accessible biomarkers in evaluating the immunonutritional status of older individuals.

## Figures and Tables

**Figure 1 nutrients-16-02464-f001:**
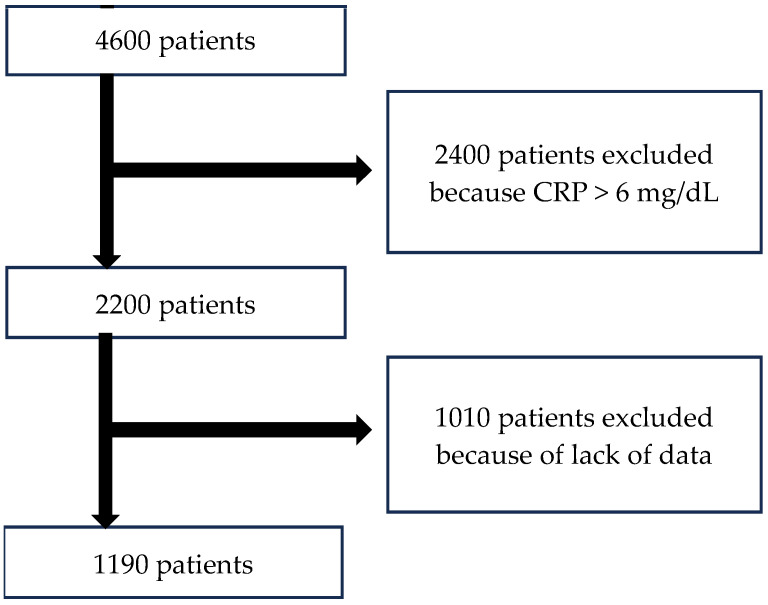
Flowchart of selecting the research group.

**Figure 2 nutrients-16-02464-f002:**
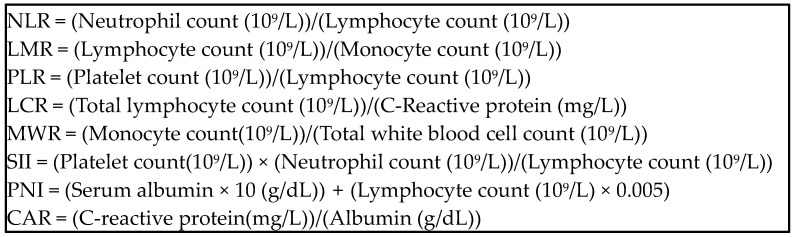
Equations of immunonutritional biomarkers.

**Figure 3 nutrients-16-02464-f003:**
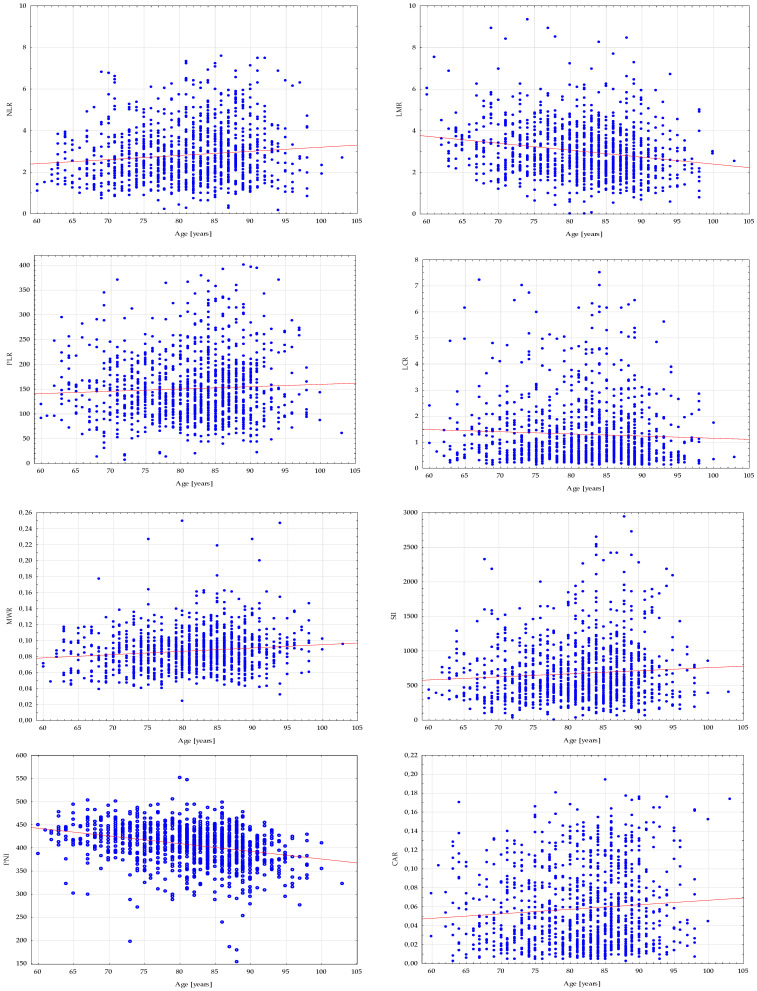
Correlation matrices for the studied biomarkers and age.

**Table 1 nutrients-16-02464-t001:** Characteristics of the subjects according to sex.

Variable	Women n = 883 mean ± SD (Median and Quartiles)	Men n = 307 Mean ± SD (Median and Quartiles)	*p*-Value
Age [years]	81.94 ± 7.98 83 (77–87)	80.79 ± 7.98 82 (75–87)	0.02
BMI [kg/m^2^]	26 ± 5.40 26 (23–30)	26.40 ± 4.27 25 (23–28)	0.39
Body mass [kg]	65 ± 14.2 64 (55–74)	76 ± 13.68 75 (67–84)	<0.001
NLR	3.15 ± 2.55 2.5 (1.8–3.6)	3.31 ± 2.69 2.7 (2.0–3.9)	0.05
LMR	3.10 ± 1.29 2.8 (2.2–3.8)	2.7 ± 1.08 2.6 (1.9–3.4)	<0.001
PLR	164 ± 93 145 (108–193)	144 ± 71 132 (98–169)	<0.001
LCR	1.3 ± 1.4 0.8 (0.4–1.6)	1.4 ± 1.5 0.8 (0.4–1.9)	0.53
MWR	0.08 ± 0.02 0.08 (0.06–0.09)	0.09 ± 0.02 0.08 (0.07–0.10)	<0.001
SII	775 ± 776 591 (391–908)	683 ± 571 524 (355–814)	0.02
PNI	406 ± 42 412 (384–433)	407 ± 41 414 (382–436)	0.66
CAR	0.05 ± 0.04 0.04 (0.02–0.08)	0.05 ± 0.04 0.04 (0.01–0.08)	0.17
Arterial hypertension; n (%)	763 (86.41%)	241 (78.50%)	0.001
Diabetes mellitus; n (%)	281 (31.86%)	115 (37.46%)	0.07
Lipid disorders; n (%)	482 (54.59%)	133 (43.32%)	<0.001
Previous stroke; n (%)	142 (16.08%)	49 (15.96%)	0.96
Coronary artery disease; n (%)	307 (34.77%)	108 (35.18%)	0.89
Previous myocardial infarction; n (%)	68 (7.70%)	48 (15.64%)	0.0005
Atrial fibrillation; n (%)	191 (21.63%)	69 (22.48%)	0.75
Heart failure; n (%)	451 (51.08%)	133 (43.32%)	0.019
Chronic kidney disease (%)	405 (45.92%)	122 (39.74%)	0.06
Obstructive lung diseases; n (%)	97 (10.99%)	35 (11.40%)	0.84
Osteoarthritis; n (%)	395 (44.73%)	98 (31.92%)	<0.001
Osteoporosis; n (%)	271 (30.69%)	38 (12.38%)	<0.001
Fractures; n (%)	144 (16.33%)	26 (8.47%)	<0.001
Gastrointestinal diseases; n (%)	200 (22.65%)	90 (29.32%)	0.019
Neoplastic diseases; n (%)	121 (13.70%)	57 (18.57%)	0.03
Depression; n (%)	386 (43.71%)	94 (30.62%)	0.001
Dementia; n (%)	361 (40.88%)	106 (34.53%)	0.04
Parkinson’s disease, n (%)	33 (3.74%)	28 (9.12%)	<0.001

Abbreviations: BMI, Body Mass Index; NLR, neutrophil-to-lymphocyte ratio; LMR, lymphocyte-to-monocyte ratio; PLR, platelet-to-lymphocyte ratio; LCR, lymphocyte-to-C-reactive protein ratio; MWR, monocyte-to-white blood cell ratio; SII, systemic immune–inflammation index; PNI, prognostic nutritional index; CAR, C-reactive protein-to-albumin ratio. Data are expressed both as the mean ± SD and median (25–75% quartiles). Variables were compared using the Mann–Whitney U-test and qualitative variables using a Chi-square test.

**Table 2 nutrients-16-02464-t002:** The correlations between immunonutritional markers and the most important quantitative variables.

Variable	Sex	Age		BMI	
		rho	*p*	rho	*p*
NLR	All	0.13	<0.001	−0.08	0.01
	Women	0.14	<0.001	−0.07	0.04
	Men	0.12	0.04	−0.10	0.09
LMR	All	−0.19	<0.001	0.09	0.004
	Women	−0.20	<0.001	0.08	0.02
	Men	−0.23	<0.001	0.11	0.06
PLR	All	0.03	0.3	−0.13	<0.001
	Women	0.05	0.14	−0.13	<0.001
	Men	−0.04	0.47	−0.15	0.01
LCR	All	−0.03	0.3	−0.15	<0.001
	Women	−0.01	0.84	−0.18	<0.001
	Men	−0.09	0.12	−0.02	0.65
MWR	All	0.11	<0.001	−0.06	0.04
	Women	0.14	<0.001	−0.06	0.08
	Men	0.07	0.23	−0.05	0.41
SII	All	0.07	0.02	−0.08	0.008
	Women	0.08	0.03	−0.07	0.039
	Men	0.03	0.61	−0.11	0.06
PNI	All	−0.28	<0.001	0.11	0.001
	Women	−0.25	<0.001	0.12	0.001
	Men	−0.35	<0.001	0.06	0.32
CAR	All	0.06	0.04	0.16	<0.001
	Women	0.04	0.30	0.19	<0.001
	Men	0.12	0.049	0.06	0.30

NLR, neutrophil-to-lymphocyte ratio; LMR, lymphocyte-to-monocyte ratio; PLR, platelet -to-lymphocyte ratio; LCR, lymphocyte-to-C-reactive protein ratio; MWR, monocyte-to-white blood cell ratio; SII, systemic immune–inflammation index; PNI, prognostic nutritional index; CAR, C-reactive protein-to-albumin ratio.

**Table 3 nutrients-16-02464-t003:** Comparison of values of immunonutritional markers between patients with and without specific diseases.

	Sex	Diabetes Mellitus	Lipid Disorder	Previous Stroke	Atrial Fibrillation	Chronic Heart Failure	Chronic Kidney Disease	Obstructive Lung Disease	Osteoarthritis	Osteoporosis	Gastrointestinal Diseases	Dementia	Parkinson’s Disease
NLR	All	↑z = 2.06	↓z = −2.47			↑z = 2.05	↑z = 2.23				↓z = −1.96		
		*p* = 0.03	*p* = 0.013	ns	ns	*p* = 0.04	*p* = 0.02	ns	ns	ns	*p* = 0.04	ns	ns
	Women			↑z = −2.33		↑z = 2.04	↑z = 1.96						
		ns	ns	*p* = 0.02	ns	*p* = 0.04	*p* = 0.04	ns	ns	ns	ns	ns	ns
	Men		↓z = −2.35			↑z = 2.05							↑z = 2.04
		ns	*p* = 0.01	ns	ns	*p* = 0.04	ns	ns	ns	ns	ns	ns	*p* = 0.04
LMR	All		↑z = 3.7	↓z = 2.04	↓z = −4.20	↓z = −2.91	↓z = −3.11	↓z = −2.46					
		ns	*p* = 0.001	*p* = 0.04	*p* < 0.001	*p* < 0.001	*p* = 0.001	*p* = 0.01	ns	ns	ns	ns	ns
	Women		↑z = 1.99	↓z = 3.02	↓z = −3.13	↓z = −3.27	↓z = −2.50	↓z = −2.46					
		ns	*p* = 0.04	*p* = 0.002	*p* = 0.001	*p* < 0.001	*p* = 0.01	*p* = 0.01	ns	ns	ns	ns	ns
	Men		↑z = 3.26		↓z = −3.09	↓z = −2.91	↓z = −2.47						
		ns	*p* = 0.001	ns	*p* = 0.001	*p* = 0.003	*p* = 0.01	ns	ns	ns	ns	ns	ns
PLR	All	↓z = −3.13			↓z = −2.04					↑z = 2.37			
		*p* = 0.001	ns	ns	*p* = 0.04	ns	ns	ns	ns	*p* < 0.001	ns	ns	ns
	Women	↓z = −2.73			↓z = −2.00								
		*p* = 0.001	ns	ns	*p* = 0.04	ns	ns	ns	ns	ns	ns	ns	ns
	Men												↑z = 2.24
		ns	ns	ns	ns	ns	ns	ns	ns	ns	ns	ns	*p* = 0.02
LCR	All		↑z = 1.96		↓z = −3.88	↓z = −2.19	↓z = −2.52			↑z = 2.36	↑z = 2.90		
		ns	*p* = 0.04	ns	*p* < 0.001	*p* < 0.03	*p* = 0.005	ns	ns	*p* < 0.01	*p* = 0.003	ns	ns
	Women				↓z = −3.41	↓z = −2.90	↓z = −2.40		↓z = −2.58	↑z = 2.54	↑z = 2.36		
		ns	ns	ns	*p* < 0.001	*p* < 0.003	*p* = 0.01	ns	*p* = 0.009	*p* < 0.01	*p* = 0.01	ns	ns
	Men		↑z = 3.09			↓z = −2.19			↑z = 2.14				
		ns	*p* = 0.001	ns	ns	*p* = 0.02	ns	ns	*p* = 0.03	ns	ns	ns	ns
MWR	All	↓z = −3.15			↑z = 4.53			↑z = 2.71			↑z = 2.19	↓z = −2.24	
		*p* = 0.001	ns	ns	*p* < 0.001	ns	ns	*p* = 0.006	ns	ns	*p* = 0.02	*p* = 0.02	ns
	Women	↓z = −3.20			↑z = 3.06	↑z = 2.32		↑z = 2.41					
		*p* < 0.001	ns	ns	*p* = 0.002	*p* = 0.02	ns	*p* = 0.01	ns	ns	ns	ns	ns
	Men				↑z = 3.84								
		ns	ns	ns	*p* < 0.001	ns	ns	ns	ns	ns	ns	ns	ns
SII	All				↓z = −2.18								
		ns	ns	ns	*p* = 0.02	ns	ns	ns	ns	ns	ns	ns	ns
	Women				↓z = −1.97								
		ns	ns	ns	*p* = 0.04	ns	ns	ns	ns	ns	ns	ns	ns
	Men												
		ns	ns	ns	ns	ns	ns	ns	ns	ns	ns	ns	ns
PNI	All		↑z = 6.69	↓z = −4.74	↓z = −5.15	↓z = −3.82	↓z = −4.19		↑z = 3.22	z		↓z = −7.28	
		ns	*p* < 0.001	*p* < 0.001	*p* < 0.001	*p* < 0.001	*p* < 0.001	ns	*p* = 0.001	ns	ns	*p* < 0.001	ns
	Women		↑z = 5.02	↓z = −4.38	↓z = −4.48	↓z = −3.60	↓z = −3.10		↑z = 2.55	↑z = 2.12		↓z = −6.24	
		ns	*p* < 0.001	*p* < 0.001	*p* < 0.001	*p* < 0.001	*p* = 0.001	ns	*p* = 0.01	*p* = 0.03	ns	*p* < 0.001	ns
	Men		↑z = −4.81		↓z = −2.55	↓z = −3.82	↓z = −2.95		↑z = 2.21			↓z = −3.71	
		ns	*p* < 0.001	ns	*p* = 0.01	*p* < 0.001	*p* = 0.003	ns	*p* = 0.02	ns		*p* < 0.001	ns
CAR	All	↑z = 3.87			↑z = 3.84	↑z = 3.504	↑z = 3.04			↓z = −3.09	↓z = −2.36		
		*p* < 0.001	ns	ns	*p* < 0.001	*p* < 0.002	*p* = 0.002	ns	ns	*p* < 0.001	*p* = 0.01	ns	ns
	Women	↑z = 3.40			↑z = 3.64	↑z = 3.30	↑z = 2.69		↑z = 2.31	↓z = −3.30			↓z = −2.08
		*p* < 0.001	ns	ns	*p* < 0.001	*p* < 0.001	*p* = 0.007	ns	*p* = 0.02	*p* < 0.001	ns	ns	*p* = 0.03
	Men	↑z = 1.99	↓z = −2.16					↑z = 2.1	↓z = −2.06			↑z = 2.04	
		*p* = 0.04	*p* = 0.03	ns	ns	ns	ns	*p* = 0.03	*p* = 0.03	ns	ns	*p* = 0.04	ns

All calculations were performed with the Mann–Whiney test. ↑—Value significantly higher in the group with a particular disease in comparison with the group without the disease. ↓—Value significantly lower in the group with a particular disease in comparison with the group without the disease.

**Table 4 nutrients-16-02464-t004:** Equations of general linear models for immunonutritional markers.

General Linear Model	R^2^	*p*
NLR = −0.63 + 0.047 × Age [years]	0.15	<0.001
LMR = 5.32 − 0.03 × Age [years] − 0.08 [if present heart failure] − 0.08 [if present atrial fibrillation] − 0.2 [if man]	0.26	<0.001
PLR = 211.75 − 2.38 × BMI [kg/m^2^] − 6.67 [if present atrial fibrillation] − 10.5 [if man]	0.18	<0.001
LCR = 2.44 − 0.04 × BMI [kg/m^2^]	0.13	<0.001
MWR = 0.06 + 0.0003 × Age [years] + 0.003 [if man] − 0.002 [if present diabetes mellitus] + 0.004 [if present atrial fibrillation] − 0.002 [if present dementia]	0.13	<0.001
SII = 1040.61 − 11.48 × BMI [kg/m^2^]	0.05	<0.001
PNI = 504.73 − 1.28 × Age [years] + 5.90 [if present lipid disorders] − 5.48 [if present previous stroke] − 3.39 [if present atrial fibrillation] − 4.95 [if present dementia] + 2.66 [ if present osteoarthritis]	0.15	<0.001
CAR = −0.013 + 0.0004 × Age [years] + 0.0012 × BMI [kg/m^2^] + 0.003 [if present diabetes mellitus] + 0.005 [if present atrial fibrillation] − 0.003 [if present osteoporosis]	0.24	<0.001

## Data Availability

The statistical data used to support presented findings may be obtained by sending a request to the corresponding author due to privacy.

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
