# Peer review of "How Immunonutritional Markers Are Associated with Age, Sex, Body Mass Index and the Most Common Chronic Diseases in the Hospitalized Geriatric Population—A Cross Sectional Study"

_nutrients, 2024, doi:10.3390/nu16152464_

Round 1

Reviewer 1 Report

Comments and Suggestions for Authors

The authors study a variety of derived biomarkers from routine haematology and chemistry blood tests in a cohort of 1190 elderly hospitalized patients. The patients are age 60 and above, admitted to the geriatric unit between 2012 and 2023 and have CRP of < 6 mg/dl. They study the biomarkers in relation to age, sex, BMI and presence of chronic diseases and find that age, sex and BMI are significantly correlated with these markers, and some of these markers are significantly different in those with specific chronic diseases vs those who do not have these diseases. The manuscript is generally well written and clearly presented. However there are some points which require clarification. 

1. In the study design 2.1, it is not clear whether and how the authors ensure that the patients who are studied are different individuals as some patients may have multiple admissions over the years. 

2. In the data section 2.2, it is not clear how the laboratory data are chosen for analyses. Many patients have repeated blood counts and CRP measurements over the course of their stay in hospital and these numbers can change over the period of days. Did the authors only choose the measurements done on admission or on discharge? 

3. It would be beneficial also to note the diagnosis for the admission during which the laboratory data were taken for analyses, as this can influence the laboratory results. 

4. In Table 3, the analyses should be adjusted for age and BMI as these also influence the laboratory data and the chronic diseases. 

5. It would be good if the average age of the patients is in the abstract.

6. The title is misleading as the cohort is not taken from the general population but rather from hospitalized patients. 

Author Response

Reviewer 1. The authors study a variety of derived biomarkers from routine haematology and chemistry blood tests in a cohort of 1190 elderly hospitalized patients. The patients are age 60 and above, admitted to the geriatric unit between 2012 and 2023 and have CRP of < 6 mg/dl. They study the biomarkers in relation to age, sex, BMI and presence of chronic diseases and find that age, sex and BMI are significantly correlated with these markers, and some of these markers are significantly different in those with specific chronic diseases vs those who do not have these diseases. The manuscript is generally well written and clearly presented. However there are some points which require clarification. 

  1. In the study design 2.1, it is not clear whether and how the authors ensure that the patients who are studied are different individuals as some patients may have multiple admissions over the years. 

Thank you for the remark. The patients who were enrolled into this analysis are different individuals. Multiple admissions of the same patients over the years were removed and only the data from first hospitalization was employed to the study. This sentence was added. “Multiple/ Consecutive admissions of the same individuals from over the years were not included in this analysis.”

  1. In the data section 2.2, it is not clear how the laboratory data are chosen for analyses. Many patients have repeated blood counts and CRP measurements over the course of their stay in hospital and these numbers can change over the period of days. Did the authors only choose the measurements done on admission or on discharge? 

Thank you for the remark. As for this study, laboratory data parameters and values were used and obtained upon admission for every patient. Section 2.2 has been amended. The sentence: “All laboratory parameters were obtained upon patients’ admission” was added.

  1. It would be beneficial also to note the diagnosis for the admission during which the laboratory data were taken for analyses, as this can influence the laboratory results.

Thank you for the comment and suggestion. First, laboratory test data were taken at admission. As we mentioned, we excluded subjects with increased CRP >= 6, to eliminate those subjects who had the infection.  This approach allowed us to obtain homogenous results in terms of inflammation.

According to the suggestion of the Reviewer, we added the sentence to the methods section:All the participants were subjects with non-infectious diseases and usually several concomitant diagnoses (e.g., heart failure, coronary heart disease, dementia, depression, anaemia, osteoporosis, chronic kidney disease, pulmonary disorders, gastrointestinal disorders, falls, fractures and diabetes mellitus).

However, full list of the disorders is much longer, that is why we do not mention all of the diseases in the main text of the paper. 

  1. In Table 3, the analyses should be adjusted for age and BMI as these also influence the laboratory data and the chronic diseases. 

Thank you for the remark. Our initial idea of preparing the table was to present the data adjusted to age and BMI as suggested by Reviewer but we resigned because our data did not show significantly the differences. Furthermore, our approach was to present data step by step. Table 1 shows bidirectional statistics in sex whereas table 2 shows for age, sex and BMI. Multivariable analysis was presented in table 4 with adjustment for age, sex, BMI and concomitant diseases. That is why we would like to leave the Table 3 in the presented form.

  1. It would be good if the average age of the patients is in the abstract.

Thank you for the comment. The mean age of the study population was stated in the result section. However, the average age (mean) of the patients were also added to the abstract.

  1. The title is misleading as the cohort is not taken from the general population but rather from hospitalized patients. 

Thank you for the comment, title has been amended from “How the Immunonutritional Markers Are Associated with Age, Sex, Body Mass Index and Most Common Chronic Diseases in Geriatric Population” to “How the Immunonutritional Markers Are Associated with Age, Sex, Body Mass Index and Most Common Chronic Diseases in Hospitalized Geriatric Population”.

Reviewer 2 Report

Comments and Suggestions for Authors

Stephenson and co-authors comprehensively examined the relationship between major chronic diseases & "immunonutritional" markers (i.e., various ratio measurements among neutrophils, lymphocytes, monocytes, platelets, C-reactive protein, & albumin) in human subjects 60+ years of age in Lodz, Poland. Remarkably, 8 immunonutritional ratios were measured against 12 age-related chronic diseases in males and females (total = 1,190 subjects). In brief, the authors found that the prognostic nutritional index [PNI; serum albumin x 10 (g/dL)) + (Lymphocyte count (109/L) × 0.005] was the strongest statistical indicator across advancing age from 60 years old, body mass index, and various chronic diseases (e.g., diabetes, dementia, atrial fibrillation). Other than thoroughly editing the manuscript again for stray typos or grammatical errors (e.g., Line 80, "...the are..."), I don't have suggestions for further improvement.

Comments on the Quality of English Language

In my opinion, the manuscript only needs another thorough editing for stray typos and grammatical errors.

Author Response

Stephenson and co-authors comprehensively examined the relationship between major chronic diseases & "immunonutritional" markers (i.e., various ratio measurements among neutrophils, lymphocytes, monocytes, platelets, C-reactive protein, & albumin) in human subjects 60+ years of age in Lodz, Poland. Remarkably, 8 immunonutritional ratios were measured against 12 age-related chronic diseases in males and females (total = 1,190 subjects). In brief, the authors found that the prognostic nutritional index [PNI; serum albumin x 10 (g/dL)) + (Lymphocyte count (109/L) × 0.005] was the strongest statistical indicator across advancing age from 60 years old, body mass index, and various chronic diseases (e.g., diabetes, dementia, atrial fibrillation). Other than thoroughly editing the manuscript again for stray typos or grammatical errors (e.g., Line 80, "...the are..."), I don't have suggestions for further improvement.

Dear Reviewer, thank you for the kind input, review and information. We agree that typos should be check in details. Regarding the correction on grammatical errors, Line 80 has been corrected and further inspection for the whole manuscript has been thoroughly done.

Reviewer 3 Report

Comments and Suggestions for Authors

This is a well-designed and well-conducted study that examines the relationship between different chronic diseases and nutritional and immune markers within a geriatric population. The authors demonstrate how chronic pathologies can alter the markers examined. The authors also show how age, BMI, and gender stand out as fundamental and crucial variables that influence the levels of nutritional and immune markers.

To be considered for publication, authors must make some minor revisions:

1. Authors must clearly define the aim of the study in the introduction.

2. The authors must indicate whether it is a cross-sectional study conducted in a geriatric population or a retrospective study.

Author Response

This is a well-designed and well-conducted study that examines the relationship between different chronic diseases and nutritional and immune markers within a geriatric population. The authors demonstrate how chronic pathologies can alter the markers examined. The authors also show how age, BMI, and gender stand out as fundamental and crucial variables that influence the levels of nutritional and immune markers.

To be considered for publication, authors must make some minor revisions:

  1. Authors must clearly define the aim of the study in the introduction.

Thank you for the kind remark. The aim was stated in the last sentence in the 4th paragraph in the introduction section. The sentence was amended to “Therefore, the aim of the current study was to explore the association of inflammatory markers (NLR, LMR, PLR, LCR, MWR, SII, PNI and CAR) with age, sex, BMI and concomitant diseases and to identify the factors potentially influencing nutritional and immunological status in older adults' hospitalized population.

  1. The authors must indicate whether it is a cross-sectional study conducted in a geriatric population or a retrospective study.

Thank you for the comment. It is a cross-sectional study. The information has been added to the title and in the material section.

Round 2

Reviewer 1 Report

Comments and Suggestions for Authors

All comments have been adequately addressed.